# The trisaccharide melezitose impacts honey bees and their intestinal microbiota

Victoria Charlotte Seeburger[1,2]*, Paul D'Alvise[2], Basel Shaaban[3], Karsten Schweikert[4], Gertrud Lohaus[3], Annette Schroeder[1], Martin Hasselmann[2]

**1** Apicultural State Institute, University of Hohenheim, Stuttgart, Baden-Wuerttemberg, Germany, **2** Department of Livestock Population Genomics, University of Hohenheim, Stuttgart, Baden-Wuerttemberg, Germany, **3** Molecular Plant Science/ Plant Biochemistry, University of Wuppertal, Wuppertal, Nordrhein-Westfalen, Germany, **4** Core Facility Hohenheim and Institute of Economics, University of Hohenheim, Stuttgart, Baden-Wuerttemberg, Germany

\* victoria.seeburger@uni-hohenheim.de

**Data Availability Statement:** The 16S amplicon sequence data generated during the current study are available from Dryad using the doi: https://doi.org/10.5061/dryad.3ffbg79f8.

## Abstract

In general, honey bees (*Apis mellifera* L.) feed on honey produced from collected nectar. In the absence of nectar, during certain times of the year or in monocultural landscapes, honey bees forage on honeydew. Honeydew is excreted by different herbivores of the order *Hemiptera* that consume phloem sap of plant species. In comparison to nectar, honeydew is composed of a higher variety of sugars and additional sugars with higher molecular weight, like the trisaccharide melezitose that can be a major constituent of honeydew. However, melezitose-containing honey is known to cause malnutrition in overwintering honey bees. Following the hypothesis that melezitose may be the cause for the so called 'honeydew flow disease', three independent feeding experiments with caged bees were conducted in consecutive years. Bees fed with melezitose showed increased food uptake, higher gut weights and elevated mortality compared to bees fed a control diet. Moreover, severe disease symptoms, such as swollen abdomen, abdomen tipping and impaired movement were observed in melezitose-fed bees. 16S-amplicon sequencing indicated that the melezitose diet changed the species composition of the lactic acid bacteria community within the gut microbiota. Based on these results, we conclude that melezitose cannot be easily digested by the host and may accumulate in the hindgut. Within cages or during winter, when there is no opportunity for excretion, the accumulated melezitose can cause severe intestinal symptoms and death of the bees, probably as result of poor melezitose metabolism capabilities in the intestinal microbiota. These findings confirm the causal relation between the trisaccharide melezitose and the honeydew flow disease and indicate a possible mechanism of pathogenesis.

## Introduction

'Honeydew flow disease' is a common problem for managed honey bee colonies. The disease occurs, when honey bees feed on honeydew honey, especially during winter [1,2]. The clinical

**Funding:** The project is supported by funds of the Federal Ministry of Food and Agriculture (BMEL) based on the decision of the Parliament of the Federal Republic of Germany via the Federal Office for Agriculture and Food (BLE) under the innovation support program to A.S. (2816500114) and G.L. (2816500214). The funders had no role in study design, data collection and analysis, decision to publish, or preparation of the manuscript. No additional external funding was received for this study. https://www.bmel.de/EN/Homepage/homepage_node.html https://www.ble.de/EN/Home/home_node.html.

**Competing interests:** The authors have declared that no competing interests exist.

symptoms of this food toxicosis are diverse: a high number of bees remain at the hive entrance instead of foraging, bees experience massive loss of hair (which may indicate complication with virus infections) and changes their behavioural patterns [3]. Even necrotic appearances could be shown in the midgut of honey bees when fed with honeydew honey in tent experiments [3]. This condition can deteriorate to a point where colonies collapse within short time. This is known to beekeepers and discussed in beekeeping journals [4], but the precise reason has remained unknown.

Honey bees primarily forage on nectar. The three most common nectar sugars are the two monosaccharides glucose and fructose, and the disaccharide sucrose [5]. However, seasonally or locally nectar plants are lacking, and nectar is not available. In such situations honey bees tend to forage on honeydew, which is not produced from nectaries, but by herbivores of the order *Hemiptera* as waste product from phloem sap of plant species [1]. In addition to the nectar sugars, honeydew contains more disaccharides such as maltose and melibiose, and trisaccharides such as erlose and melezitose [6]. These sugars are produced from aphids with α-glucosidase/transglucosidase [7].

Honey bees process nectar into blossom honey and honeydew into honeydew honey. In comparison to blossom honey, the mineral content (aluminium, boron, copper, magnesium, manganese, nickel and zinc) of honeydew honeys is up to four times higher [8,9]. Moreover, honeydew honey contains more oligosaccharides than blossom honey [10]. Oligosaccharides are known to be poorly assimilated by honey bees and lead to increased losses of winter colonies fed on honeydew honey [11]. In earlier literature, the reason for these losses were discussed, and it has been assumed that this specific kind of honey impacts wintering colonies because of anatomical effects on the bee gut, microbial changes or restricted assimilation of nutrients, which leads to a higher mortality rate [11]. Following the hypothesis that the increased mineral contents of honeydew honeys could be the reason for the honeydew flow disease, feeding experiments with sugar solutions of different mineral contents were conducted [3]. However, the typical honeydew flow disease symptoms could not be produced, even though physiological damages in the midgut of bees fed with sugar solutions with higher mineral content occurred [3]. Despite these arising concerns about honeydew and the processed honey, the exact reason for this honeydew flow disease symptoms are not identified yet.

In the present study we followed the hypothesis that honeydew flow disease could be caused by the trisaccharide melezitose. Melezitose is the primary trisaccharide in honeydew, especially in the more common honeydew of aphids that live on spruces, where it can constitute up to 70% of the sugar fraction [6]. This trisaccharide is composed of two glucose and one fructose molecules [7]. It is not clear whether honey bees can digest melezitose; other hymenopteran species are known to have the ability to process melezitose. Many studies discuss the preference of ants for melezitose [12–17] and their capability for digesting melezitose is established [18]. Before it can pass the intestinal epithelium, melezitose has to be metabolised to hexose units by specific enzymes. The link between a fructose and a glucose unit can be broken down by invertase [18], which is commonly present in the lumen of the ant gut [19]. Also under laboratory conditions, it was observed that invertase can break down melezitose [20]. Invertase can be detected in varying concentrations in honey bees. The concentration of invertase increases with the age of the bees in summer bees and is constantly present in winter bees in high concentration [21]. Since honey bees are also known to possess invertase [21,22], it is possible, that they are capable of breaking down melezitose in the same way as ants. In experiments in which sucrose, maltose, melezitose and trehalose were fed to honey bees, their melezitose metabolites glucose and fructose appeared in the haemolymph. However, melezitose did also appear in the haemolymph as unmetabolised molecule, while sucrose and maltose were metabolised to a greater extent [23]. Caged bees prefer sucrose over all other sugars,

namely: arabinose, xylose, fructose, glucose, galactose, mannose, lactose, maltose, melibiose, trehalose, raffinose and melezitose [24].

Further knowledge on possible dietary and health effects can be gained by characterising the intestinal microbiota, which consists of only few species in honey bees [25–31]. Since mono- and oligosaccharides constitute the main energy source for honey bees [24,32], components of sugar uptake systems, such as phosphotransferase system family genes, are enriched in their gut microbiota [28]. Genes for sugar transport and carbohydrate breakdown are enriched specifically in the microbial species *Gilliamella apicola*, *Bifidobacterium* spp., and the *Lactobacillus* species clusters Firm-4 and Firm-5 [27,28,33]. Especially the carbohydrate-degrading enzymes found in *Gilliamella apicola*, *Lactobacillus* Firm-4, *Lactobacillus* Firm-5 and *Bifidobacterium* spp. are beneficial for breaking down nectar sugars to use them as energy sources [27]. The bacteria digest carbohydrates and produce short-chain fatty acids as fermentation products. Honey bees can utilise both the sugars and their fermentation products for energy metabolism [34]. The gut microbiota could be important for degradation of more complex sugars that may otherwise have toxic effects [27].

Honeydew with high contents of the trisaccharide melezitose could cause the severe clinical symptoms of the honeydew flow disease in honey bees due to poor melezitose metabolism capabilities of the intestinal microbiota. To understand the impact of melezitose on honey bee health, feeding experiments with melezitose were performed, during which physiological condition and behaviour of the bees were monitored, and the intestinal microbiota was analysed by 16S-amplicon sequencing.

## Material and methods

### Performance of the feeding experiments

Four feeding experiments were performed during summer of the years 2017, 2018 and 2019. European honey bees (*Apis mellifera*) were collected from the hives of the Apicultural State Institute (University of Hohenheim, Germany). For every experiment, six brood combs without adult bees were removed from three different donor colonies, caged and incubated at 33˚C for 24 h. Newly emerged bees (day 1) were collected and pooled. Out of the pooled bees, 50 bees were randomly collected and placed in each one of twelve experimental cages. Bees were held in cages, as previously described [35]. The cages were placed in a darkened climate chamber in randomised block design at the typical brood nest temperature of 35˚C. Melezitose does not crystallise at this temperature. The bees were fed *ad libitum* with control feed (39% (w/v) fructose, 31% glucose, 30% sucrose) or melezitose feed (50% melezitose, 19.5% fructose, 15.5% glucose), which mimicked the sugar spectrum of honeydew honey with high melezitose content.

Six cages were supplied with control feed and six cages with melezitose feed. Both sugar solutions were treated in the same way, dissolved in an ultrasonic bath that heats up in 30 min from 23˚C to 70˚C. 2 ml of the respective sugar solution was freshly prepared daily to ensure same viscosity and no effect on the degree of crystallisation. Solutions were administered simultaneously with vials in each cage until all bees had died.

**Feeding experiment 1 – Sugar analyses of crop content.** In the first cage experiment, all bees were frozen at day 21. This was necessary to for crop content analysis in order to prove the ability of all bees to collect and process the food solutions with different sugar compositions. The crop content, if present, was collected for sugar analysis (34 crops of control-fed bees and 69 crops of melezitose-fed bees). The sugars were analysed according to Lohaus and Schwerdtfeger [36].

**Feeding experiment 2 –Sugar analyses of processed sugar solution.** In the second cage experiment, bees of all six melezitose-fed cages collected the sugar solution in small honey combs that they built from the provided wax foundation rectangle. The sugar proportions of the collected, processed feed were analysed [36] for each cage (six processed sugar solutions).

**Feeding experiment 3 –Analyses of processed sugar solution, gut microbiota and water supply.** Again, the bees in two of the melezitose-fed cages collected the sugar solution in honey combs, and the sugar proportions were analysed [36] for each cage (two processed sugar solutions).

In the third cage experiment, we performed a preliminary gut microbiome analysis. At day 10, one live bee was collected from one control-fed cage, and two live bees from two different melezitose-fed cages. At day 15 and 20 this procedure was repeated. The collected bees (N = 9) were frozen immediately at -80˚C. Further description of the preliminary gut microbiota analysis is provided below.

In the third cage experiment, distilled water was supplied *ad libitum* in centrifuge tubes; in addition to the sugar feed solution.

**Feeding experiment 4 –Analyses of gut microbiota.** In the fourth cage experiment, gut microbiota analysis was implemented. The results of the preliminary microbiota analysis in cage experiment three indicated that for acquisition of a complete gut microbiota, the caged bees needed contact to nurse bees [37,38]. To ensure this, the experimental design was adjusted. 1'832 newly emerged bees from the donor colonies described above (day 1) were marked with liquid water-proof marker in a colour representing their colony (1, 2 and 3). Later, on the same day, the marked bees were placed back into their donor colonies. Four days later (day 5), 20 marked bees from each donor colony were placed in each of the nine experimental cages. Six cages were fed with the control solution and three cages with the melezitose solution. Moreover, from day 10 on, three of the six control-fed cages were changed to melezitose diet ("changed diet").

For the gut microbiota analysis, bees were collected at different time points. On day 5, six bees from each donor colony (3 donor colonies x 6 bees = 18 bees) were collected. On day 10, six bees were collected from each of the three control and three melezitose-fed cages. Constantly two of the collected bees per cage originated from donor colony 1, two from donor colony 2 and two from donor colony 3, as identified by their colour marks (2 treatment groups x 3 cages x 3 donor colonies x 2 bees = 36 bees). On day 15, six bees were collected from each cage (3 treatment groups x 3 cages x 3 donor colonies x 2 bees = 54 bees). The collected bees (N = 108) were frozen immediately at -80˚C. Further procedures in the gut microbiota analysis are described below.

## Additional field experiments

Additionally, at five sampling sites in the black forest (Southern Germany), 100 bees from three bee colonies per sampling site were collected during honeydew season in 2017 and 2018. The crop contents from each hive were pooled for sugar analysis (5 sampling sites x 3 colonies x 2 years = 30 analyses).

## Analysis of the aspects of honey bee health

In order to measure the aspects of bee health (see Table 1) in feeding experiments 2–4, food uptake per cage was recorded daily by weighing of the food [g]. The food uptake was then calculated difference to the food weight given the day before. Mortality was recorded by counting the dead bees exactly every 24 hours. The whole body weights without crop and the weights of the dissected guts of the first ten dead bees in each cage were recorded.

**Table 1. Overview of the analyses as well as the number of bees and cages per treatment of the feeding experiments performed between 2017 and 2019.**

| Feeding experiment | Year | Sugar analyses | Microbiota analysis | Aspects of bee health | Treatment groups | Cages per treatment | Bees per cage |
|---|---|---|---|---|---|---|---|
| 1 | 2017 | Cf, Cc, Fw | / | / | Co, Me | 6 | 50 |
| 2 | 2017 | Cc, Fw | / | FU, GW, ST | Co, Me | 6 | 50 |
| 3 | 2018 | / | 9 | FU, GW, ST | Co, Me | 6 | 50+3 |
| 4 | 2019 | / | 108 | FU, GW, ST | Co, Me, M10 | 3 | 60+12 |

The sugar content was analysed in crops of bees in the field (Cf), crops of bees from the feeding experiments (Cc) and from feed the bees transported into the cells of the beeswax foundation rectangle in the cages (Fw). Aspects of bee health monitored by daily food uptake per cage (FU), gut-body weight ratio of dead bees (GW) and survival of all bees per cage (ST). The treatment groups were fed a control solution of sucrose, glucose and fructose (Co) or with a 1:1 solution of control and melezitose (Me), in 2019 control-fed bees were fed with melezitose from day 10 (M10). The extra bees for microbiota analysis were marked and put in the cages additionally (noted with +).

## Gut microbiota profiling

To profile the gut microbiota, DNA of nine bees from the feeding experiment in 2018, as well as from 108 bees from the feeding experiment in 2019 was extracted using a TRIzol protocol. Whole single bees were extracted using our standard protocol because DNA and RNA can be extracted simultaneously and be used for further experiments. The bees were placed in a 2 ml lysis tube with five 0.8 mm steel beads, roughly 50 μl 0.1 mm glass/zirconia beads and 0.5 ml TRIzol (Invitrogen). The bees were homogenised on a FastPrep24 (MP Bio) at 5.5 m/s for 50 s. After 5 min of incubation at RT, 100 μl chloroform was added and the contents were mixed by vigorous shaking, followed by 5 min of incubation at RT. The two phases were separated by 15 min centrifugation at 12.000 $g$ and 4˚C. The aqueous phase was transferred to another tube for RNA extraction. 250 μl back extraction buffer (4 M guanidine thio-cyanate, 50 mM sodium citrate, 1 M TRIS base) was added to the rest of the homogenate and mixed by vigorous shaking. After 10 min of incubation at RT and centrifugation for 15 min at 12.000 $g$ and 4˚C, the aqueous phase was transferred to a new tube with 200 μl isopropanol and mixed by repeated inverting. After 5 min of incubation at RT and 15 min of centrifugation (12.000 $g$, 4˚C), the supernatant was removed, the pellet was washed with 80% ethanol, dried for 10 min at RT and centrifuged again (12.000 $g$, 4˚C) for 5 min. The supernatant was removed, the pellet dried for 5 min at RT and redissolved in 50 μl 8mM NAOH. After another centrifugation for 10 min (12.000 $g$, RT) to remove the membrane lipids, the supernatant was transferred into 4.25 μl 0.1 M HEPES and 0.5 μl RNAse A (Amresco 10 mg/ml), mixed carefully and incubated for 1 h at 37˚C. DNA concentrations were determined using Qubit fluorometer (Thermo Fisher Scientific). The resulting DNA concentrations ranged between 10.1–94.6 ng/μl. Amplicons from the V3-V4 region of the 16S-rRNA-gene were generated and Illumina-sequenced using 20 ng template DNA (Eurofins Genomics, Ebersberg, Germany). The PCR conditions, library preparation, sequencing and initial data preparation were described previously [39]. After demultiplexing by demultiplexor_v3.pl (Perl 5.30) and initial quality filtering, OTU binning (97% identity) was done by USEARCH 8.0 [40], as well as quality filtering and Chimera filtering by UCHIME [41] (with RDP set 15 as a reference database). The sequencing data were analysed on the Integrated Microbial NGS platform [42], using a 0.1% total abundance threshold. This is a UPARSE based analysis pipeline reporting OTU sequences with ≤1% incorrect bases in artificial microbial community tests [43]. Primary taxonomic classification was done by RDP classifier version 2.11 training set 15 [44] and sequence alignment was done by MUSCLE [45]. The taxonomic classification was controlled and refined by BLAST-searching the representative OTU sequences in the NCBI database (https://blast.ncbi.nlm.nih.gov).

Normalisation, taxonomic binning, and statistical analyses were carried out using the RHEA scripts [46] on R studio version 1.1.456.

## Ethics statement

In accordance with the guidelines of the authors' institutions' and the applicable regulations, no ethics approval was required or obtained for the present study. This study was carried out in Baden-Wuerttemberg, Germany. Honey bees are no subjects of the German Animal Protection law. Additionally, neither endangered nor protected species were involved in this study.

## Statistical analyses

The daily food uptake per bee was calculated in consideration of the number of bees alive on the respective day. The gut-body weight ratio was calculated from the weight of the recorded bee bodies (without crop) and their removed guts. In order to visualise the results for both measures, box plots were created for each group in the respective year. A linear regression was used to estimate the group differences in daily food uptake controlling for the number of bees alive, daily and annual effects. Since the gut-body weight ratio range between 0 and 1, a fractional logit regression model was employed to estimate group differences controlling for year effects and the age of bees. Survival of bees was analysed in a Cox proportional hazards model. Differences between bacterial species in the treatment groups were analysed with the Rhea R pipeline [46]. All statistical tests were conducted and graphs were drawn using R version 3.5.2.

**Table 2. Results of the linear regression model (daily food uptake), fractional logit regression model (gut-body weight ratio) and Cox proportional hazard model (survival).**

| Coefficients | Estimate | St. Error | Pr(>|t|) |
|---|---|---|---|
| **Panel I: Daily food uptake** | | | |
| Intercept | 0.1162 | 0.0096 | <0.001 |
| Melezitose | 0.0027 | 0.0029 | 0.947 |
| Melezitose day +10 | 0.0184 | 0.0061 | 0.003 |
| 2018 | 0.0109 | 0.0030 | <0.001 |
| 2019 | 0.0151 | 0.0040 | <0.001 |
| Days | -0.0018 | 0.0003 | <0.001 |
| Bees_alive | -0.0017 | 0.0001 | <0.001 |
| **Panel II: Gut-body weight ratio** | | | |
| Intercept | -0.0520 | 0.1099 | 0.636 |
| Melezitose | 0.0866 | 0.0154 | <0.001 |
| Melezitose day +10 | 0.0550 | 0.0260 | 0.034 |
| 2018 | 0.0157 | 0.0136 | 0.248 |
| 2019 | 0.0028 | 0.0142 | 0.842 |
| Days | 0.0012 | 0.0016 | 0.440 |
| **Panel III: Survival** | | | |
| Melezitose | 0.9319 | 0.0555 | <0.001 |
| Melezitose day +10 | 2.2514 | 0.1073 | <0.001 |
| 2018 | -0.1144 | 0.0584 | 0.050 |
| 2019 | -0.8595 | 0.0771 | <0.001 |

The multivariate $R^2$ for the linear regression in Panel I is 0.253 and the F-test for overall significance yields F(6.893) = 50.4 (p < 0.001). For the fractional logit regression in Panel II, we report average partial effects. The multivariate $R^2$ is 0.174 based on 330 observations. The log-rank test (4.1740) yields 628.1 (p < 0.001).

## Results

The bees fed with melezitose in the experiments showed disease symptoms related to their physiological condition and generic behaviour. Bees fed with the control diet stayed predominantly on the beeswax foundation rectangle (S1 Video) and bees fed with melezitose were observed to crawl mostly on the bottom of the cage (S2 Video). In fact, towards the end of the experiment, this behaviour was exhibited more frequently. Additionally, melezitose-fed bees also moved more often and faster than the bees in the control-fed cages. Swollen abdomens, abdomen tipping, impaired movement, twitching and terminal paralysis were observed during all feeding experiments in the melezitose-fed bees (S2 Video and S3 Video).

Following our hypothesis that melezitose affects the health of honey bees, we conducted several multivariate regression analyses investigating multiple aspects of honey bee health. Using daily food uptake per bee, gut-body weight ratio and survival time as dependent variables, we can show that melezitose has a highly significant negative effect on honey bee health. The results of our individual regression models are presented in Table 2.

### Sugar analyses of crop contents in cage and field experiments and of processed feed

The results of the sugar spectrum analysis in the crops of the caged bees point out that the trisaccharide melezitose was taken up and degraded into the small molecule sugars trehalose, sucrose, glucose and fructose. While the melezitose proportion in the food was 50%, the mean value in the crops from 69 bees was 18.88%.

The sugar analyses of the contents from the field and laboratory experiments showed that the bees ingested melezitose. Presence of liquid in the crop indicated active feeding. The crops of the field-collected bees contained up to 10.8% melezitose. Besides melezitose, these crops contained (in ranking order) mainly glucose and fructose, less than 10% sucrose, trehalose, turanose, maltose and erlose. Furthermore, less than 1% consisted of melibiose, raffinose and stachyose.

The mean melezitose content in the processed feed in the combs formed from the beeswax foundation rectangles in the cages was 28.92% (8 processed sugar solutions of melezitose-fed cages) (Table 3).

### Effects of melezitose feeding on water and food uptake

Bees did not take up more water when fed with melezitose feed, as compared to control solution and also did not show a significantly higher food uptake than control-fed bees (Fig 1). In contrast, the food uptake of bees fed with melezitose starting from day 10 was 20 mg higher ($p < 0.01$) than that of bees fed with the control solution and higher than that of bees fed with melezitose from the first day onwards ($p < 0.01$). To illustrate the relative increase in food uptake, we determined the average uptake per bee, which was 20 mg for control-fed bees. Calculating from this value, a change from control to melezitose diet caused the bees to approximately double their food intake (Fig 1).

### Effects of melezitose feeding on viscosity and weight of guts

The proportion of gut weight of the respective bee bodies was 52% in control-fed bees, 60% in melezitose-fed bees and 56% in bees fed with melezitose from day 10 (Fig 2). We conducted a fractional logit regression ($R^2(5,330) = 0.174$) and found that both groups fed with melezitose had significantly higher gut-body weight ratios than the control group ($p < 0.001$ for bees fed with melezitose from day 1 and $p < 0.05$ for bees fed with melezitose from day 10). The effect

**Table 3. Fructose, glucose, sucrose and melezitose proportion of the feed solution, crop content of honey bees collected from the first feeding experiment, field experiment and the processed melezitose feed.**

| Analysed solution | Fructose [%] | Glucose [%] | Sucrose [%] | Melezitose [%] |
|---|---|---|---|---|
| Food solution | 19.50 | 15.50 | 15.00 | 50.00 |
| Crop content of feeding experiment (N = 69) | 38.82 | 36.04 | 6.10 | 18.88 |
| Crop content of field experiment (N = 15) | 45.30 | 35.50 | 4.60 | 3.70 |
| Processed sugar solution (N = 8) | 24.71 | 37.59 | 8.79 | 28.92 |

of the higher gut-body weight ratio was numerically slightly weaker (six percentage-points increase of gut-body-weight ratio instead of nine percentage points increase) for bees fed with melezitose starting from day 10, which implies that a longer exposure to melezitose seems to enlarge the gut of bees.

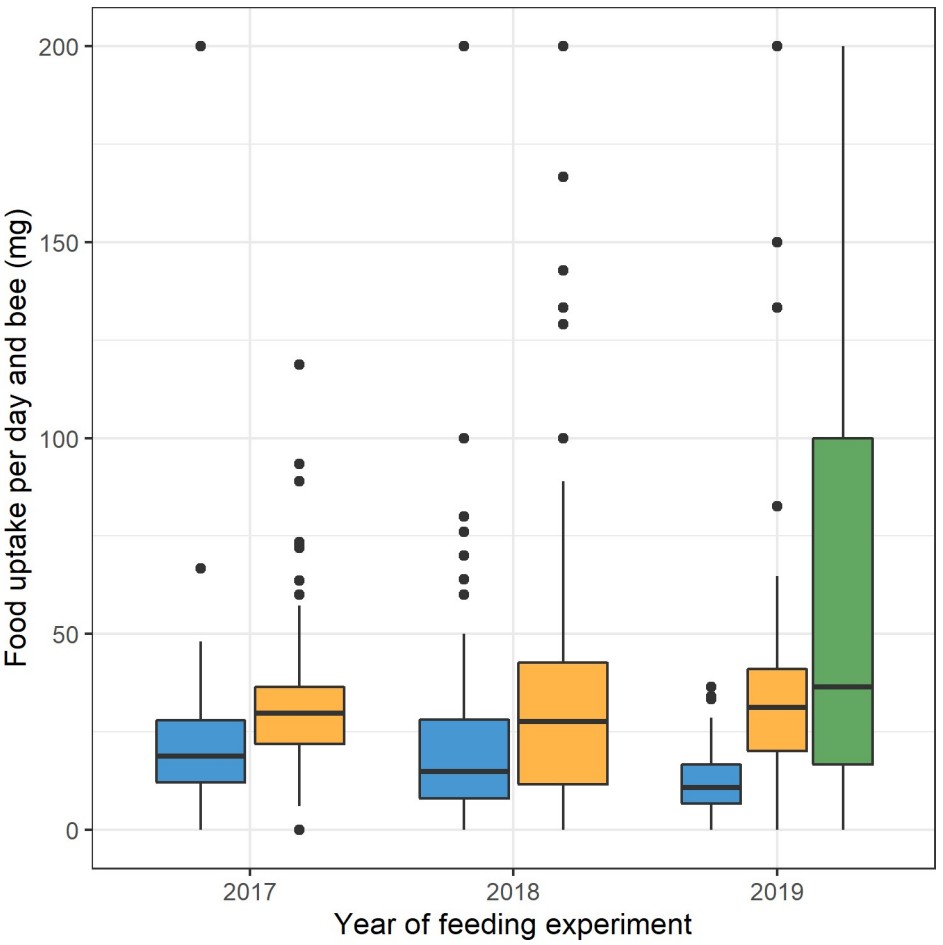

**Fig 1. Boxplot of food uptake per day and bee in milligram (mg) in all cage experiments.** The three treatment groups are highlighted with different colours. The mean value of daily food uptake per bee fed with control feed (blue) was 20 mg per day (n = 445) and 37 mg per day (n = 388) for bees fed with melezitose feed (yellow) and 70 mg per day (n = 67) for bees fed with melezitose feed from day 10 (green) of all years. The vertical boxplots depict the interquartile range (lower bound/ upper bound of the box correspond to the 25/75% quantile), the median (horizontal line in the box) and outlying observations (points outside the box).

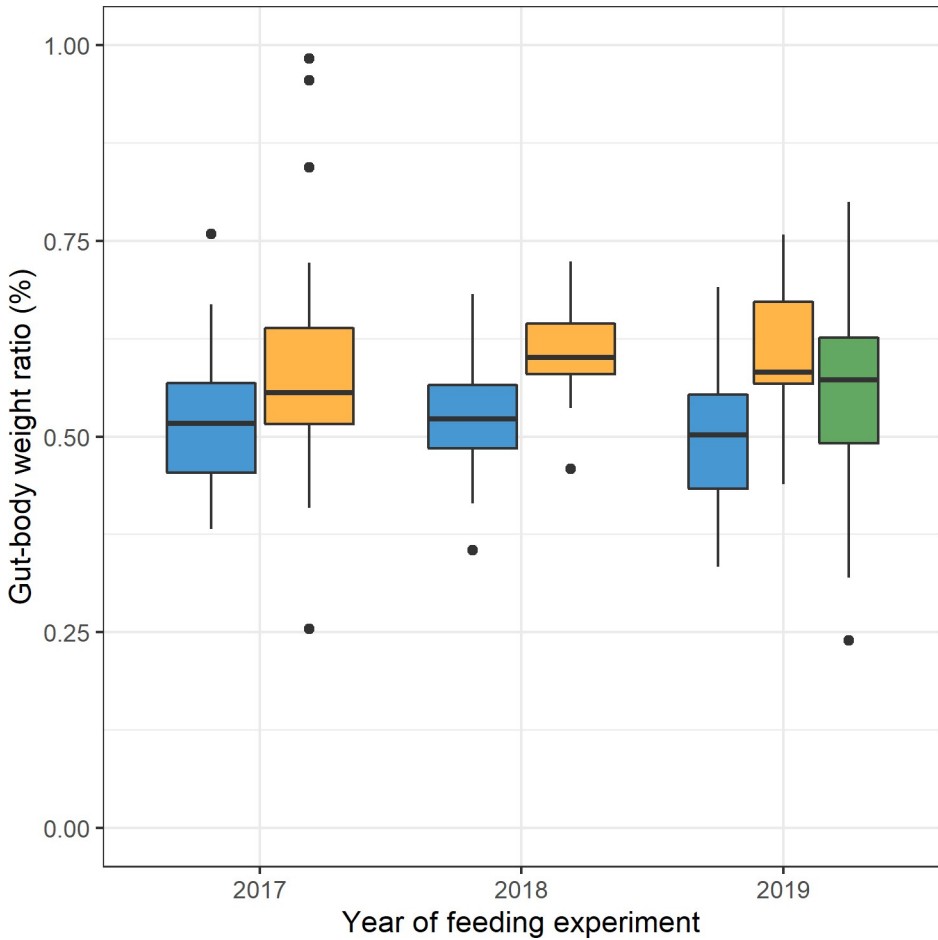

**Fig 2. Boxplot of gut-body weight ratio of honey bees in all cage experiments.** The three treatment groups are highlighted with different colours. The mean value of gut-body weight ratio per bee fed with control feed (blue) was 52% (n = 150) and 60% (n = 150) for bees fed with melezitose feed (yellow) and 56% (n = 30) for bees fed with melezitose feed from day 10 (green). The vertical boxplots depict the interquartile range (lower bound/ upper bound of the box correspond to the 25/75% quantile), the median (horizontal line in the box) and outlying observations (points outside the box).

## Effects of melezitose on survival

Estimated survival times for all feeding experiments are depicted in Fig 3. In general, the survival time of bees fed with melezitose was lower compared to the survival time of bees fed with the control solution in each year (p < 0.001). We found no significant differences between the feeding experiments in 2017 and 2018. However, the median survival rates in 2019 were significantly higher than those in 2017 and 2018 (p < 0.001). In the control group of the feeding experiment 2019, 50% of the bees had died after 29 days, while 50% of the melezitose-fed bees had died already after 25 days. Interestingly, the bees fed with melezitose starting from day 10 showed a more rapid inset of mortality than the bees fed with melezitose from the start of the experiment (p < 0.001). Their median survival time was only 17 days.

## Effects of melezitose on the gut microbiota

A preliminary, small scale (N = 9) microbiota analysis was conducted in the second experiment, to determine if there were any effects of the treatments on the microbiota. Surprisingly,

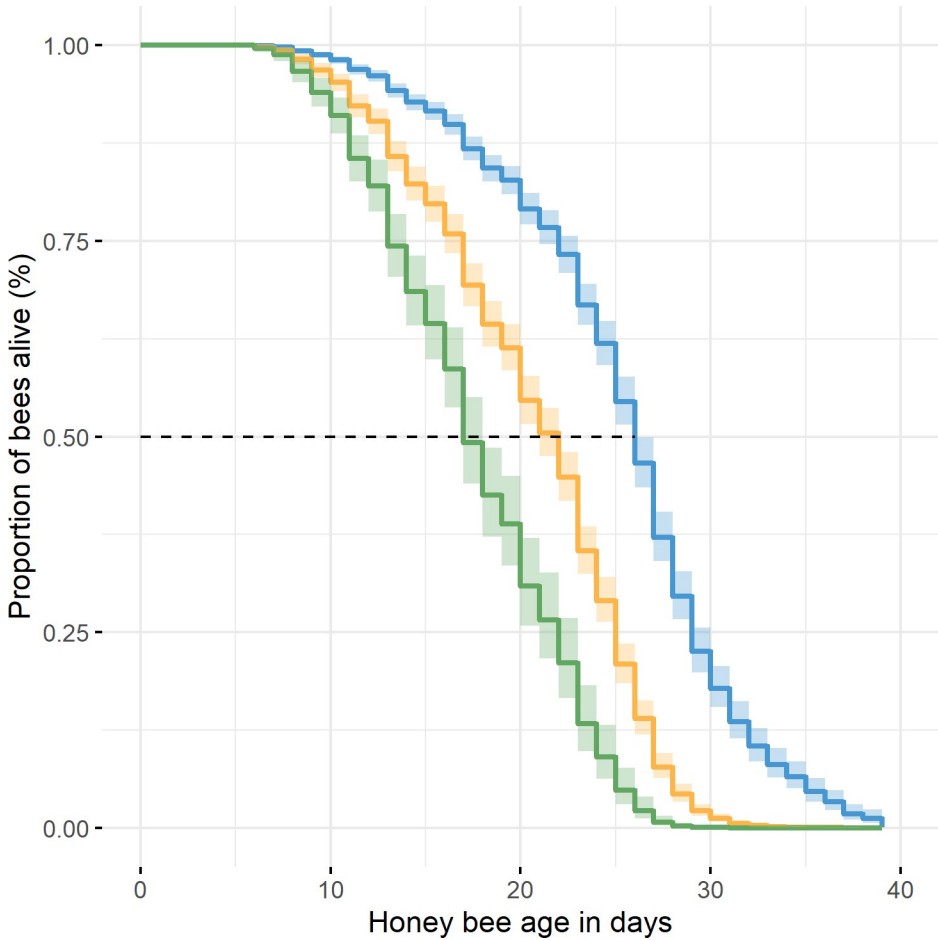

**Fig 3. Overall survival probability plots in all cage experiments.** The survival rate of the honey bees in the control (blue), in the melezitose group (yellow) and in the bees fed with melezitose from day 10 (green) (Cox regression, Log-rank (4,1740): 628.1, p < 0.001).

we found that the core microbiota members *Gilliamella apicola* and *Snodgrassella alvi* were absent from all bees analysed (S1 Fig). This is remarkable, as they are usually, together with the *Lactobacillus* species, the most common and persistent bacteria in free-living adult bees. This absence of two core-microbiota bacteria was attributed to the circumstance that the bees used in the first three experiments had emerged from their cells and were put into the cages without contact to adult nest members. Therefore, it was decided to modify the experimental setup for the third experiment in a way that the age-standardised bees come into contact with their adult nest members for five days to allow natural development of the intestinal microbiota (S2 Fig). Diversity analyses of the 16S-amplicon sequence data revealed no significant differences in the community composition between the treatment groups of the feeding experiment (S3 Fig). In the bees from the fourth experiment (N = 108) a total of ten gut bacteria species could be detected. The five core members are *Snodgrassella alvi*, *Gilliamella apicola*, *Bifidobacterium* spp., *Lactobacillus* Firm-4 and *Lactobacillus* Firm-5, as well as bacteria from the species *Frischella perrara*, *Gluconacetobacter* spp., *Parasaccharibacter apium*, *Bartonella apis* and *Lactobacillus kunkeei* (S4 Fig).

At first glance, the microbial community appeared to be unaffected by the melezitose treatment. However, on closer examination of the OTU composition, we found that the

proportions of the lactic acid bacteria differed between the treatment groups. *Lactobacillus kunkeei* increased over time in bees fed with the control diet, but was not present at all in bees fed with melezitose, and it was very low in bees fed with melezitose from day 10 (F-statistic: 2.66 on 5 and 102 DF, p = 0.03). Also, the relative abundance of *Lactobacillus* Firm-4 increased in control-fed bees and decreased in bees fed with melezitose from day 10 (Fig 5; F-statistic: 4.245 on 5 and 102 DF, p = 0.002). In contrast, the relative abundance of *Lactobacillus* Firm-5 decreased in bees fed with the control diet, but increased significantly in bees fed with melezitose and bees fed with melezitose from day 10 (Fig 6; F-statistic: 7.048 on 5 and 102 DF, p < 0.001).

## Discussion

This study describes the multiple effects of melezitose in honey bees and indicates its key significance for the occurrence of the described honeydew flow disease. This disease already led to colony losses during winter [3], which are usually noticed by beekeepers and therefore documented in beekeepers' journals [4]. In three feeding experiments, bees fed with melezitose showed intestinal symptoms, and increased food uptake, gut weight and mortality.

In order to analyse the progress of uptake and digestion of melezitose, the relative proportion of melezitose in the feed and in the crops was measured. The melezitose proportion decreased from feed to crop (Table 3), suggesting that bees or their crop microbiota did metabolise melezitose. In social insects, the proventriculus provides the individual with the amount of food needed to cover their actual energy needs, leaving as much as possible in the crop [47]. Nevertheless, honey bees are known to digest harmful food to conserve the health of the colony [48]. Thus, it can be assumed, that the individual honey bees foraging on honeydew will digest as much of the harmful melezitose as possible. This ensures that the remaining colony is provided with easily digestible food which is processed into honey.

Interestingly, we found that bees fed with melezitose from day 10 had twice as much food uptake than control-fed bees (Fig 1). Both the doubled food uptake and the increase of melezitose can lead to an accumulation of food in the gut. The average gut-body weight ratio that was eight percent higher in bees fed with melezitose (Fig 2) also explains the morphological symptom of the swollen abdomen. These results lead us to the assumption that bees need more time to digest melezitose or are unable to digest the absorbed melezitose and thus the sugar content in the intestine increases. The rising amount of melezitose can lead to the severe symptoms that were observed in these feeding experiments. Concentrating on the life expectancy of bees, the symptoms of honeydew flow disease appeared with increasing age (first on day 10 after emergence). A gradual accumulation of melezitose with the lifetime of honey bees can be assumed. The shorter lifespan of bees fed with melezitose compared to the control group could be explained by their digestive problems and their influence on the physiology of their abdomen (Fig 3).

Consequently, we expected changes in their intestinal microbiota and performed 16S-amplicon sequencing to check for microbial shifts. There was a significant shift in the lactic acid bacteria species: *Lactobacillus kunkeei* did not occur in the bees fed with melezitose (Fig 4) and the proportion of *Lactobacillus* Firm-4 decreased (Fig 5). Conversely, the proportion of *Lactobacillus* Firm-5 increased with feeding on melezitose (Fig 6). *Lactobacillus* species ferment sugars to produce lactic or acetic acid and are adapted to sugar-rich environments with high acidity [29]. They are known to be dominant in the crop and most abundant in the ileum and rectum [37,38]. Within the bee-associated *Lactobacilli*, *Lactobacillus mellifera* and *L. mellis* form a distinct phylogenetic cluster referred as *Lactobacillus* Firm-4, and the species *Lactobacillus apis*, *L helsingborgensis*, *L. kimbladii*, *L. kullabergensis*, and *L. melliventris* are referred as

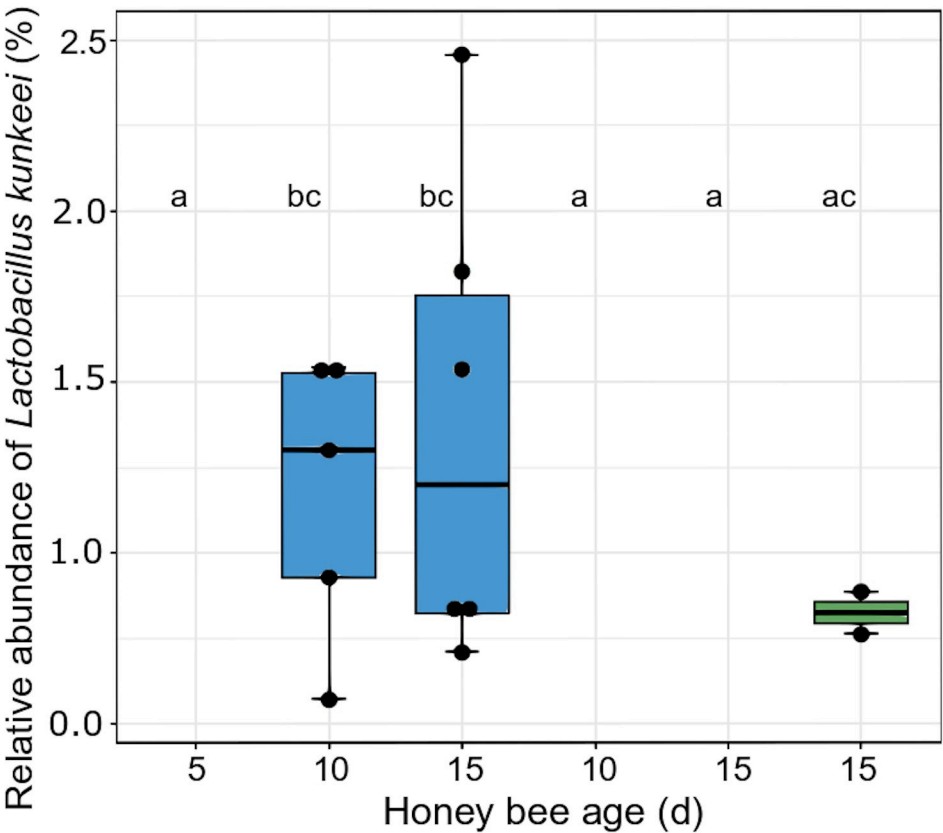

**Fig 4. Relative abundance of *Lactobacillus kunkeei* in the gut microbiota.** The three treatment groups are highlighted with different colours: control (blue), melezitose (not detected at all time points) and melezitose from day 10 (green). Significantly different groups are highlighted by the letters a, b and c (F-statistic: 2.66 on 5 and 102 DF, p = 0.027). For each treatment group and each bee age, 18 individuals were used for analysis. Not detected bacteria are marked with ND. The vertical boxplots depict the interquartile range (lower bound/ upper bound of the box correspond to the 25%/ 75% quantile), the median (horizontal line in the box) and outlying observations (points outside the box).

*Lactobacillus* Firm-5 [49]. Both clusters are located within the hindgut, *Lactobacillus* Firm-4 in the rectum, *L.* Firm-5 in the ileum and rectum [49,50]. While these two clusters are rarely detected outside the hindgut, *Lactobacillus kunkeei* is also found outside the honey bee body in the hive. *L. mellifera*, which belongs to the Firm-4 cluster is only capable of producing acids from fructose, while the species of the Firm-5 cluster can also utilise the sugars galactose, mannose, sorbose and sucrose [49]. The more diverse capabilities for oligosaccharide metabolism of *Lactobacillus* Firm-5 species may explain their increase within the melezitose-fed bees.

These findings point out the importance of lactic acid bacteria for the nutrition of their host. The bees that were fed with control diet first and from day 10 on with melezitose diet died earlier than those fed with melezitose from the fifth day onwards (Fig 3). This may be seen as further evidence for the key role of an adapted microbiota in the processing of oligosaccharides. Bees fed with melezitose from the fifth day may have grown an adapted intestinal microbiota capable of degrading the oligosaccharides at an increased rate. The change in diet from the control to melezitose diet on day 10 day may therefore have led to a rapid accumulation of melezitose in the guts of unadapted bees with acute, often lethal effects.

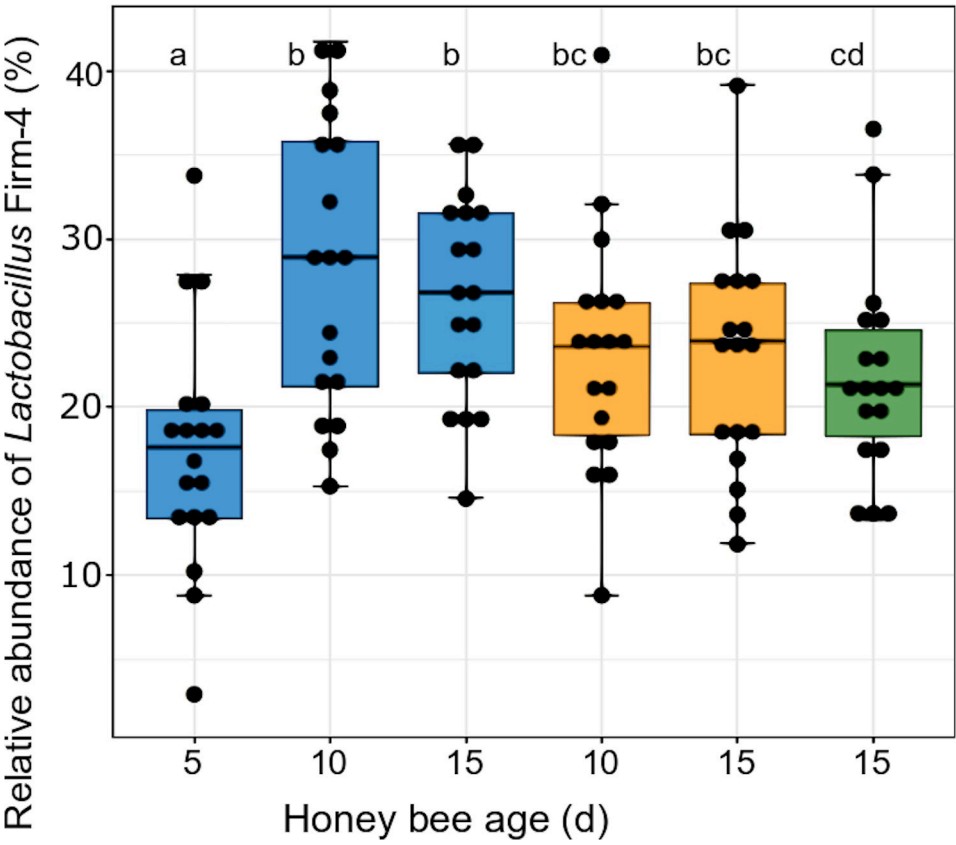

**Fig 5. Relative abundance of *Lactobacillus* Firm-4 in the gut microbiota.** The three treatment groups are highlighted with different colours: control (blue), melezitose (yellow) and melezitose from day 10 (green). Significantly different groups are highlighted by the letters a, b, c and d (F-statistic: 4.245 on 5 and 102 DF, p = 0.002). For each treatment group and each bee age, 18 individuals were used for analysis. The vertical boxplots depict the interquartile range (lower bound/ upper bound of the box correspond to the 25%/ 75% quantile), the median (horizontal line in the box) and outlying observations (points outside the box).

Honey with high proportion of melezitose is well-known to crystallise rapidly, often already within the hives [1]. However, intestinal epithelial lesions, which might be caused by uptake of melezitose crystals, are an unlikely cause of the observed symptoms, since crystallisation did not occur under the experimental conditions. No crystals were found by microscopy during the examination of the affected intestine. Another possible effect of melezitose might be dehydration. However, the bees of the third experiment fed with melezitose diet did not ingest more water than the control group; therefore dehydration can be ruled out as major cause of these symptoms. It is well-known that oligosaccharides are poorly utilised by the gut microbiota of honey bees and beekeepers should therefore try to avoid them as food source for their colonies [33,51]. Here, we to show this effect for the oligosaccharide melezitose, which is very common in honeydew of spruce forests. The shift of the lactic acid bacteria in bees fed with melezitose provides evidence for the bees' struggle to digest melezitose. On the one hand, honey bees used to live in forests, so that it can be assumed that their bacteria have had sufficient evolution time for adapting to typical honeydew sugars. On the other hand, it should be acknowledged that natural forests with their higher botanical diversity used to provide more nectar than most forests today. The occurrence of melezitose at high concentrations in the colonies is a problem primarily caused by the beekeepers themselves. Bee colonies are, on

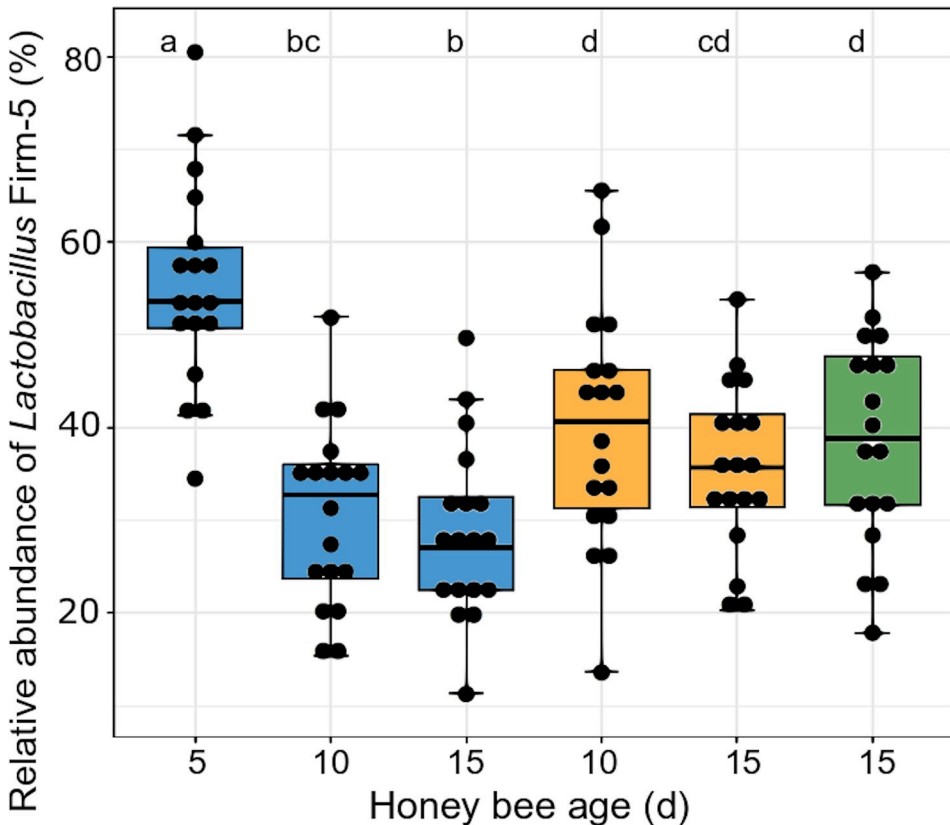

**Fig 6. Relative abundance of *Lactobacillus* Firm-5 in the gut microbiota.** The three treatment groups are highlighted with different colours: control (blue), melezitose (yellow) and melezitose from day 10 (green). Significantly different groups are highlighted by the letters a, b, c and d (F-statistic: 7.048 on 5 and 102 DF, p < 0.001). For each treatment group and bee age, 18 honey bee individuals were used for analysis. The vertical boxplots depict the interquartile range (lower bound/ upper bound of the box correspond to the 25%/ 75% quantile), the median (horizontal line in the box) and outlying observations (points outside the box).

purpose, relocated to forests with lack of nectar plants, because honeydew honey with its strong malty-aromatic taste is very popular and has a high market value [9]. The situation of bees in the cages of the feeding experiment is similar to that of bees in winter that cannot leave the hive because of low outside air temperature and therefore cannot defecate [52]. Since bees have a well-studied hygienic behaviour [53] and only defecate outside of the colony, the melezitose inevitably remains in the hindgut. The gut microbiota of honey bees may be somewhat capable of processing melezitose, however this process may take time and melezitose may accumulate in the gut faster than it can be processed. Therefore, the melezitose remains in the bee and leads to the typical symptoms of the disease. This explains the occurrence of honeydew flow disease symptoms especially during winter. Since honey bees have a longer lifespan of several months in winter instead of several weeks [54], they are more dependent on effective digestion of the stored food, which may favour honeydew flow disease in winter.

Honeydew flow disease is a regional phenomenon, mainly noticed by beekeepers in Germany, Austria and Switzerland, which can be explained by the coincidence of sufficiently cold winters, large fir-dominated forests, and beekeepers aiming to produce honeydew honey. However, wild-living honey bee colonies that are not maintained by beekeepers can be found in forests all over Europe [55] and can come in contact with melezitose, too. It can be assumed,

that these wild honey bee colonies have adapted their behaviour to honeydew with high amounts of melezitose. They probably decrease the melezitose amount in their stored honey by intensely foraging for nectar. Apart from that, the proportion of melezitose in the excretion of honeydew producer species is variable [6] and there can be regional differences.

It should be noted that other oligosaccharides such as erlose and raffinose may probably cause similar problems as melezitose. Melezitose is the most common trisaccharide in the honeydew of spruces, as the spruce is the host tree of most honeydew producer species in Europe when pines are not present locally [56]. Nevertheless, the results of these experiments can be used as an example and be transferred to the effects of other oligosaccharides. Therefore, feeding experiments with other oligosaccharides would extend the knowledge gained from this study. Further experiments with several different sugar solutions in each cage could show the preference of bees, and a transfer of the experiments to the field would give deeper insights into the effects on colony level to qualitatively different nutrition. Additionally, it should be further investigated whether and how the bee's hindgut is physiologically affected by feeding with oligosaccharides.

From the experience gained during this study in respect of establishment of a complete, natural microbiota in experimental bees, we recommend to allow for contact between the newly emerged experimental bees and nurse bees. Gut microbiota bacteria are acquired by newly emerged bees by oral-oral transmission (especially lactic acid bacteria) and by faecal-oral transmission through contact with other worker bees or hive material [37,38].

Altogether, the results of this study lead to the conclusion that melezitose affects the health of honey bees. The trisaccharide may accumulate in the gut over time as the gut microbiota needs more time to metabolise melezitose than for simple sugars. The present results show that high amounts of melezitose play a key role in the occurrence of the honeydew flow disease in bee colonies. Additionally, it can be assumed that the honeydew flow disease can affect honey bees synergistically with pathogens, such as the paralysis viruses, which are most abundant during honeydew season [57]. Bee colonies are superorganisms and can compensate diseases through healthy new brood and maintenance of homeostasis [58]. However, if melezitose accumulates in the combs and is not removed by the beekeeper, the bees will face digestion problems during the winter when no brood can be reared. Based on the results of this present study, it can be recommended to avoid honeydew with high contents of melezitose. Beekeepers should therefore remove their colonies from the forests, when environmental conditions favour melezitose production.

## Supporting information

**S1 Fig. Overall gut microbiota of sequenced DNA of honey bees from the third feeding experiment (mean value of N = 9 analysed gut microbial communities).** Honey bees were removed from their colonies directly after emerging and had no contact to nurse bees. Bacterial species are highlighted by colour and shown in the legend. Core-members of the honey bee gut microbiota are written in bold.
(TIF)

**S2 Fig. Overall gut microbiota of sequenced DNA of honey bees from the fourth feeding experiment (mean value of analysed gut microbial communities of N = 108 bees).** Honey bees lived in bee hive colonies with contact to nurse bees until day 5. Bacterial species are highlighted by colour and shown in the legend. Core-members of the honey bee gut microbiota are written in bold.
(TIF)

**S3 Fig. Differences in α-diversity, i.e. Shannon effective number of species, in the gut microbiota of control (blue) and melezitose (yellow) fed bees and bees with a changed diet (green) based on 16S RNA gene amplicon sequencing.** No significant differences between the groups were detected (Kruskal-Wallis chi-squared = 1.8162, df = 5, p-value = 0.8739). For each treatment group and honey bee age, 18 honey bee individuals were used for analysis. C = control-fed bees (blue), M = melezitose-fed bees (yellow), CM = bees first fed with control and from day 10 with melezitose (green); 5, 10 and 15 shows the honey bee age in days. The vertical boxplots depict the interquartile range (lower bound/ upper bound of the box correspond to the 25%/ 75% quantile) and the median (horizontal line in the box). (TIFF)

**S4 Fig. Absolute abundances of ten microbiota members monitored in the feeding experiment 2019.** Absolute abundance of the ten monitored phylotypes: *Snodgrassella alvi*, *Gilliamella apicola*, *Bifidobacterium* spp., *Lactobacillus* Firm-4, *Lactobacillus* Firm-5, *Frischella perrara*, *Gluconacetobacter* spp., *Parasaccharibacter apium*, *Bartonella apis* and *Lactobacillus kunkeei*. The ten plots show the cumulative abundances for each bee. For each treatment group, 18 honey bee individuals were used for analysis. C = control-fed bees (blue), M = melezitose-fed bees (yellow), CM = bees first fed with control and from day 10 with melezitose (green); 5, 10 and 15 shows the honey bee age in days. Significant differences between the treatment groups could be shown for all *Lactobacillus* species and are demonstrated in Figs 4–6. The vertical boxplots depict the interquartile range (lower bound/ upper bound of the box correspond to the 25%/ 75% quantile), the median (horizontal line in the box) and outlying observations (points outside the box). (TIFF)

**S1 Video. Control-fed bees of one cage of feeding experiment three.** The bees fed with control solution predominantly stayed on the beeswax foundation rectangle and moved only little. (MP4)

**S2 Video. Melezitose-fed bees of one cage of feeding experiment three.** Bees fed with melezitose were observed to mostly crawl on the bottom of the cage and moved very often and fast. They displayed the disease symptoms: swollen abdomen, abdomen tipping, impaired movement, twitching and terminal paralysis. (MP4)

**S3 Video. Melezitose-fed bees of one cage at the end of feeding experiment three.** Bees fed with melezitose showed swollen abdomens and impaired movements, which was more severe towards the ends of their live. Eventually, they are unable move and succumb to the disease. (MP4)

## Author Contributions

**Conceptualization:** Victoria Charlotte Seeburger, Paul D'Alvise, Annette Schroeder, Martin Hasselmann.

**Data curation:** Victoria Charlotte Seeburger, Paul D'Alvise, Basel Shaaban, Karsten Schweikert.

**Formal analysis:** Victoria Charlotte Seeburger, Paul D'Alvise, Karsten Schweikert.

**Funding acquisition:** Basel Shaaban, Gertrud Lohaus, Annette Schroeder.

**Methodology:** Victoria Charlotte Seeburger, Paul D'Alvise, Basel Shaaban, Gertrud Lohaus, Annette Schroeder, Martin Hasselmann.

**Project administration:** Victoria Charlotte Seeburger.

**Validation:** Victoria Charlotte Seeburger, Paul D'Alvise, Martin Hasselmann.

**Visualization:** Victoria Charlotte Seeburger, Paul D'Alvise, Karsten Schweikert.

**Writing – original draft:** Victoria Charlotte Seeburger.

**Writing – review & editing:** Victoria Charlotte Seeburger, Paul D'Alvise, Basel Shaaban, Karsten Schweikert, Gertrud Lohaus, Annette Schroeder, Martin Hasselmann.

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
