## [Decision Letter · Decision Letter 0]

6 Feb 2020

PONE-D-19-34528

The trisaccharide melezitose impacts honey bees and their intestinal microbiota

PLOS ONE

Dear Mrs Seeburger,

Thank you for submitting your manuscript to PLOS ONE. After careful consideration, we feel that it has merit but does not fully meet PLOS ONE’s publication criteria as it currently stands. Therefore, we invite you to submit a revised version of the manuscript that addresses the points raised during the review process. All reviewers agree that this is a valuable study, but I concur that its impact will be further increased by correcting some of the language issues.

We would appreciate receiving your revised manuscript by Mar 22 2020 11:59PM. To enhance the reproducibility of your results, we recommend that if applicable you deposit your laboratory protocols in protocols.io, where a protocol can be assigned its own identifier (DOI) such that it can be cited independently in the future. For instructions see: http://journals.plos.org/plosone/s/submission-guidelines#loc-laboratory-protocols

We look forward to receiving your revised manuscript.

Kind regards,

Olav Rueppell

Academic Editor

PLOS ONE

Journal Requirements:

2. Please include in your methods section the full species name of the bees and also the source of the bees used in this study.

4. Please remove your figures from within your manuscript file, leaving only the individual TIFF/EPS image files, uploaded separately.  These will be automatically included in the reviewers’ PDF.

Reviewers' comments:

Reviewer's Responses to Questions

**Comments to the Author**

1. Is the manuscript technically sound, and do the data support the conclusions?

Reviewer #1: Yes

Reviewer #2: Yes

Reviewer #3: Yes

2. Has the statistical analysis been performed appropriately and rigorously? 

Reviewer #1: Yes

Reviewer #2: I Don't Know

Reviewer #3: Yes

3. Have the authors made all data underlying the findings in their manuscript fully available?

Reviewer #1: Yes

Reviewer #2: Yes

Reviewer #3: Yes

4. Is the manuscript presented in an intelligible fashion and written in standard English?

Reviewer #1: Yes

Reviewer #2: Yes

Reviewer #3: Yes

5. Review Comments to the Author

Reviewer #1: The manuscript by Seeburger et al. describes important experiments on the action of the trisaccharide melezitose on honeybees. In the absence of nectar, honeybees frequently collect honeydew which consists of herbivore excretions. Honey containing honeydew contains large amounts of melezitose and is known to cause malnutrition in overwintering honeybees. The authors performed very interesting feeding experiments with caged bees and show that melezitose-fed bees increased food uptake, had higher gut weighs and elevated mortality compared to control bees. Also, they showed swollen abdomen and impaired movements. Melezitose further changed their lactic acid bacteria community. The authors conclude that melezitose cannot be digested easily and may accumulate in the hindgut, thereby causing severe intestinal symptoms.

The topic of this article is important and exciting for the basic and applied honeybee research community. The manuscript is written very well. Methods and results are explained in a clear way. The statistics are performed correctly. The results are discussed in detail. However, English language could be improved and the authors are advised to have an English native speaker correct their manuscript.

Otherwise, I only have a couple of minor points:

1) l. 80 „. . . The concentration of invertase . . . in summer bees and occurs is constantly present in winter bees consistently “; something went wrong with this sentence, please correct.

2) l. 135. Please give a rationale for freezing bees at day 21

3) l. 270, please correct „as was the case”

4) Fig. 1 Consider using mg instead of g

5) Fig. ff. What do the error bars indicate? SEM? STD?

6) l .357 “fed with the control diet”

7) l. 386, correct “led to colony losses”

8) l. 387 “beekeepers’ journals”

9) l. 420, correct “died earlier than those fed with . . . .”

10) the authors might want to discuss in more detail why wild honeybee colonies in forests (not only in Germany but world-wide) apparently do not suffer as much from honeydew than honeybee colonies maintained by beekeepers. Is there an adaptation to melezitose possible?

Reviewer #2: Overall comments:

1. This is an interesting study, and well-structured, with useful and relevant information and excellent findings. That melezitose influences honey bee health is made very clear, and links to microbiome-mediated changes are being suggested.

2. There are very minor English issues, in terms of spelling, grammar, which may be corrected during the editor check. For e.g. ad libitum should be in italics, no need for a “-“ between number and units, consistency of units to SI, centrifugal force g in italics, for the reagents the company name to be followed by city and country, melezitose feds hould have a hyphen in between, spp. not to be in italics, Firm-4 an Firm 5 are usually not in italics., Lactobacillus should always be in italics,

Detailed comments:

Abstract

Higher molecular weight?

Digested by the host?

Rephrase 16S-amplicon sequencing?

Introduction

Line 53. Mineral content higher by how much? If it is 2 % it may not matter, if it is 20 % it may.

Line 187. Why were whole bees used? Why not just the gut pulled out?

Results

Line 285/286. Perhaps delete one of the “on the other hand”…. And rephrase.

Line 305. What is numerically slightly weaker?

Discussion

Lines 394-396. Perhaps rephrase. It is not clear what is being meant here.

Line 414- 418. The shift from themicrobiome to a lower Firm-4/L. kunkeei and higher Firm-5 relative abundance may be better explained by describing the metabolic versatility and location of the bacteria that are seen in these two groups. Please refer to https://www.ncbi.nlm.nih.gov/pubmed/24944337 and https://www.ncbi.nlm.nih.gov/pmc/articles/PMC6478020/

And that much better explains your statements in lines 421-423.

Line 432. What does “sugar drains the bees“ mean? Not clear.

Line 435- “sever problems” is too general. Please describe in terms of viability, or metabolic changes or changes in populations.

Line 460. Compensate what problems?

Line 465-467. The reason for this assumption is not clear at all.

Reviewer #3: The authors Seeburger et al. report melezitose-containing honey effects on honey bees. Melezitose is shown to be the cause for the so called ‘honeydew flow disease'. Three independent feeding experiments with caged bees as well as field bees were examined. Bees fed with melezitose showed intake effects, gut weights effects, and showed a lower survival. Several disease symptoms were recorded in melezitose-fed bees. Melezitose changed the gut microbiota too. The MS is written in a clear and concise manner. The Title and Abstract cover the MS contents. The Introduction gives a strong basis. A need for this study is clear.

This MS gives valuable insight to what extent melezitose evokes honey bee colony disease behaviors. The M&M are clear and described into appropriate depth. The statistics are applied appropriately too. Results are reported clearly, and also properly covered by the Discussion. I think the figures are well selected, regarding placement of many as Supplementary Materials. Reference are good, with no inappropriate citations. The conclusion, that melezitose cannot be easily digested and may accumulate in the hindgut is supported by the results. Accumulated melezitose causes intestinal symptoms from which the bees can die. Overall, I like to complement the authors with this MS which is a pleasure to read.

6. PLOS authors have the option to publish the peer review history of their article (what does this mean?). If published, this will include your full peer review and any attached files.

Reviewer #1: No

Reviewer #2: No

Reviewer #3: No

---

## [Author Response · Author response to Decision Letter 0]

9 Mar 2020

Reviewer's Responses to Questions

Comments to the Author

1. Is the manuscript technically sound, and do the data support the conclusions?

Reviewer #1: Yes

Reviewer #2: Yes

Reviewer #3: Yes

2. Has the statistical analysis been performed appropriately and rigorously? 

Reviewer #1: Yes

Reviewer #2: I Don't Know

Reviewer #3: Yes

3. Have the authors made all data underlying the findings in their manuscript fully available?

Reviewer #1: Yes

Reviewer #2: Yes

Reviewer #3: Yes

4. Is the manuscript presented in an intelligible fashion and written in standard English?

Reviewer #1: Yes

Reviewer #2: Yes

Reviewer #3: Yes

5. Review Comments to the Author

Reviewer #1: The manuscript by Seeburger et al. describes important experiments on the action of the trisaccharide melezitose on honeybees. In the absence of nectar, honeybees frequently collect honeydew which consists of herbivore excretions. Honey containing honeydew contains large amounts of melezitose and is known to cause malnutrition in overwintering honeybees. The authors performed very interesting feeding experiments with caged bees and show that melezitose-fed bees increased food uptake, had higher gut weighs and elevated mortality compared to control bees. Also, they showed swollen abdomen and impaired movements. Melezitose further changed their lactic acid bacteria community. The authors conclude that melezitose cannot be digested easily and may accumulate in the hindgut, thereby causing severe intestinal symptoms.

The topic of this article is important and exciting for the basic and applied honeybee research community. The manuscript is written very well. Methods and results are explained in a clear way. The statistics are performed correctly. The results are discussed in detail. However, English language could be improved and the authors are advised to have an English native speaker correct their manuscript.

Thank you for your review and comments. We carefully improved the language throughout the manuscript.

Otherwise, I only have a couple of minor points:

1) l. 80 „. . . The concentration of invertase . . . in summer bees and occurs is constantly present in winter bees consistently “; something went wrong with this sentence, please correct.

We corrected this sentence (L. 84).

2) l. 135. Please give a rationale for freezing bees at day 21

The reason is added. We needed to freeze the bees for proofing the ability of honey bees to collect and process the food solution with the different sugar compositions in general (L. 133-135).

3) l. 270, please correct „as was the case”

We changed the phrase (L. 280-281).

4) Fig. 1 Consider using mg instead of g

We changed Fig. 1 accordingly o the comment.

5) Fig. ff. What do the error bars indicate? SEM? STD?

The bars in the boxplot highlight the 25%/75% quantiles (lower bound/ upper bound). They help us to visualise and compare the distributions displayed in Figure 1, 2, 4-6, S3 and S4. We added this information to the figure legends.

6) l .357 “fed with the control diet”

We corrected this sentence (L. 368).

7) l. 386, correct “led to colony losses”

We corrected this sentence (L. 398).

8) l. 387 “beekeepers’ journals”

We corrected this (L. 399).

9) l. 420, correct “died earlier than those fed with . . . .”

We corrected this (L. 444-445).

10) the authors might want to discuss in more detail why wild honeybee colonies in forests (not only in Germany but world-wide) apparently do not suffer as much from honeydew than honeybee colonies maintained by beekeepers. Is there an adaptation to melezitose possible?

Thank you, we discussed this in line 483-492.

Reviewer #2: Overall comments:

1. This is an interesting study, and well-structured, with useful and relevant information and excellent findings. That melezitose influences honey bee health is made very clear, and links to microbiome-mediated changes are being suggested.

Thank you very much for your comments and valuable suggestions.

2. There are very minor English issues, in terms of spelling, grammar, which may be corrected during the editor check. For e.g. ad libitum should be in italics, no need for a “-“ between number and units, consistency of units to SI, centrifugal force g in italics, for the reagents the company name to be followed by city and country, melezitose feds hould have a hyphen in between, spp. not to be in italics, Firm-4 an Firm 5 are usually not in italics., Lactobacillus should always be in italics,

We improved the English language throughout the manuscript and added the missing information.

Detailed comments:

Abstract

Higher molecular weight?

We changed this (L. 20).

Digested by the host?

We also made this change (L.31).

Rephrase 16S-amplicon sequencing?

We rephrased it (L. 28-30).

Introduction

Line 53. Mineral content higher by how much? If it is 2 % it may not matter, if it is 20 % it may.

We noticed a second publication reporting this exactly. The mineral content of honeydew honeys compared to blossom honeys is, dependent on the mineral, up to four times higher (L. 55-58).

Line 187. Why were whole bees used? Why not just the gut pulled out?

We are using the RNA-DNA-Extraction as a standard protocol, where whole bees are used and the RNA and DNA can be extracted simultaneously (L.199-200), suitable for follow-up questions.

Results

Line 285/286. Perhaps delete one of the “on the other hand”…. And rephrase.

We rephrased this (L. 296-298).

Line 305. What is numerically slightly weaker?

The effect of the higher gut-body weight ratio was numerically weaker (six percentage-points increase of gut-body weight ratio instead of nine percentage points increase) for the bees fed with melezitose starting from day ten. We added this to the text (L. 316-318).

Discussion

Lines 394-396. Perhaps rephrase. It is not clear what is being meant here.

We rephrased it (L. 408-411).

Line 414- 418. The shift from the microbiome to a lower Firm-4/L. kunkeei and higher Firm-5 relative abundance may be better explained by describing the metabolic versatility and location of the bacteria that are seen in these two groups. Please refer to https://www.ncbi.nlm.nih.gov/pubmed/24944337 and https://www.ncbi.nlm.nih.gov/pmc/articles/PMC6478020/

And that much better explains your statements in lines 421-423.

Thank you very much for suggesting these references, we included them (L. 432-442).

Line 432. What does “sugar drains the bees“ mean? Not clear.

We rephrased it with ‘dehydration’ (L. 457).

Line 435- “sever problems” is too general. Please describe in terms of viability, or metabolic changes or changes in populations.

We explained that melezitose is poorly utilised by honey bees (L. 459-461).

Line 460. Compensate what problems?

We referred to honey bee diseases in general (L. 516).

Line 465-467. The reason for this assumption is not clear at all.

We rephrased this and added a reference (L. 513-515).

Reviewer #3: The authors Seeburger et al. report melezitose-containing honey effects on honey bees. Melezitose is shown to be the cause for the so called ‘honeydew flow disease'. Three independent feeding experiments with caged bees as well as field bees were examined. Bees fed with melezitose showed intake effects, gut weights effects, and showed a lower survival. Several disease symptoms were recorded in melezitose-fed bees. Melezitose changed the gut microbiota too. The MS is written in a clear and concise manner. The Title and Abstract cover the MS contents. The Introduction gives a strong basis. A need for this study is clear.

This MS gives valuable insight to what extent melezitose evokes honey bee colony disease behaviors. The M&M are clear and described into appropriate depth. The statistics are applied appropriately too. Results are reported clearly, and also properly covered by the Discussion. I think the figures are well selected, regarding placement of many as Supplementary Materials. Reference are good, with no inappropriate citations. The conclusion, that melezitose cannot be easily digested and may accumulate in the hindgut is supported by the results. Accumulated melezitose causes intestinal symptoms from which the bees can die. Overall, I like to complement the authors with this MS which is a pleasure to read.

 Thank you for sharing your opinion on our manuscript and the valuable feedback. 

6. PLOS authors have the option to publish the peer review history of their article (what does this mean?). If published, this will include your full peer review and any attached files.

Do you want your identity to be public for this peer review? For information about this choice, including consent withdrawal, please see our Privacy Policy.

Reviewer #1: No

Reviewer #2: No

Reviewer #3: No

---

## [Editor Report · Decision Letter 1]

11 Mar 2020

The trisaccharide melezitose impacts honey bees and their intestinal microbiota

PONE-D-19-34528R1

Dear Dr. Seeburger,

We are pleased to inform you that your manuscript has been judged scientifically suitable for publication and will be formally accepted for publication once it complies with all outstanding technical requirements.

With kind regards,

Olav Rueppell

Academic Editor

PLOS ONE
---

## [Editor Report · Acceptance letter]

26 Mar 2020

PONE-D-19-34528R1 

The trisaccharide melezitose impacts honey bees and their intestinal microbiota 

Dear Dr. Seeburger:

I am pleased to inform you that your manuscript has been deemed suitable for publication in PLOS ONE. Congratulations! Your manuscript is now with our production department. 

With kind regards,

on behalf of

Dr. Olav Rueppell 

Academic Editor

PLOS ONE